# Implementing a multi-cycle datapath with Liquid Marbles

**Sandro Erba\*, Luca Cavenaghi, Claudio Zandron**

Dipartimento di Informatica, Sistemistica e Comunicazione, Università degli Studi di Milano-Bicocca, Milan, Italy

\* s.erba9@campus.unimib.it

## Abstract

Liquid Marbles are liquid droplets encapsulated by hydrophobic powder particles; due to their non-wetting nature, they allow to manipulate liquids efficiently. Literature highlighted their potential to be employed as micro-reactors, micro-containers for growing micro-organisms and cells, micro-fluidics devices, and have also been used in the framework of unconventional computing. In this work, we discuss a theoretical implementation of all required components to define a multi-cycle datapath based on Liquid Marbles. Then, we consider issues related to scalability, by discussing how the circuits can be expanded with the growth of the inputs, and also how they can be modified to overcome the issues related to the growing time and space complexity.

**Data Availability Statement:** All relevant data are within the manuscript.

**Funding:** Ministero Università e Ricerca of Italy, under the grant "Dipartimenti di Eccellenza 2023-2027" of the Department of Informatics, Systems

## Introduction

Various proposals for implementing unconventional models of computation have appeared in the scientific literature. These include glider-based computing, which originates from Conway's Game of Life [1, 2], chemical reactions such as the Belousov-Zhabotinsky reaction [3–5], or Cellular Automata [6, 7], just to cite a few examples.

Another such a paradigm recently appeared deals with a special type of liquid based computing [8], which make use of liquid marbles. Liquid Marbles (LMs), first introduced in 2001 [9], represent liquid droplets enveloped by hydrophobic powders. These marbles exhibit a combination of elastic and liquid properties, that have been explored in various studies [10–12]. For example, Jin et al. [13] proposed an energy relationship to determine the coalescence condition of LMs. Coalescence, signifying the merging of LMs upon collision, occurs when kinetic energy surpasses the surface energy of an individual LM. The Weber number, defined as $We = \frac{rDv^2}{s}$, captures the relationship between kinetic and surface energy.

In the work by Jin et al. [13], it was calculated that for coalescence to occur, the kinetic energy of an individual LM should be at least around 60% of its initial surface energy. Under this energy limit, the LMs exhibit elastic rebounds and follow altered trajectories [14]. Properties of Liquid Marbles, including the optimal casing composition (Ni/UHDPE) for electromagnet control [15], were analyzed in [16, 17]. The ability to control LMs through electromagnets

and Communication of the University of Milano-Bicocca, Italy.

**Competing interests:** NO authors have competing interests.

is crucial for developing a computing system based on LMs, where precise timing is essential to ensure proper system functioning.

The computing model based on Liquid Marbles (LMs) proposed in [15] is similar to the collision-based Billiard Ball Model (BBM) [18]. However, LMs exhibit behavior that aligns with the Soft Sphere Model (SSM), particularly exemplified by the Margolus gate [18]. The primary distinction of this model lies in the arrangement of paths that collect the marbles at the output. The management of merged LMs is addressed using a hydrophobic scalpel [19]. The computational potential of the interaction gate, serving as a *half adder*, is demonstrated [15]. This logic is further extended to implement a one-bit *full adder* and other circuits, including an actuator [20].

The goal of our research is to build on these works, to propose a possible, and purely theoretical, implementation of a multi-cycle datapath based primarily on computation via liquid marbles. Throughout the paper, combinational and sequential circuits are presented that are based on the liquid marble interaction gate [21]. Although a possible implementation of various logic gates with LMs has already been discussed [22], directly exploiting such gates to schematize a multi-cycle datapath using classical electronic device schemes would have entailed too many problems, such as excessive LMs loss, complex routing issues, and the requirement for circuits of a significative size. A direct design of such circuits is thus proposed, discussing the implementation of each component of a datapath by making use of LM logic, and discussing scalability, approximate size, and possible problems arising from the use of LMs. The correctness of the proposed solutions has been verified by means of the BBM simulator presented in [23].

The rest of the paper is organized as follows. In Section we present some guidelines for the correct reading of the proposed diagrams used throughout the paper, and a general schema of the multi-cycle datapath we are going to deal with. In Section we present the description of some simplest circuits, namely those for *multiplexers* and *decoders*, which will be widely used in the following steps to build upon the rest of the circuits. In Section the implementation of sequential circuits for saving and exchanging data and instructions will be presented, while the device to execute instruction, the *ALU*, will be presented in Section. Section describes the *control logic* circuits, used to control the entire multi-cycle datapath. The paper ends with some discussions in Section and some conclusions in Section.

## Overview

The first proposal for a logic gate based on LMs was presented in [15], where a collision gate that resembles the collision-based BBM [18] was discussed. The implementation of this logic gate was based on with the interaction gate presented in [21]. In this case, the input LMs are run on two ramps facing each other. The meeting of LMs can generate different results, and to collect them all, five output channels are required, each with a different angle. The two outermost channels will only contain an LM if there is only one LM between the two inputs. The three innermost channels will contain LMs if and only if there is an interaction between the two input LMs. For these results to be meaningful, the LMs must be correctly synchronised throughout the circuit.

As explained in [21], the collision between two LMs can either generate a new LM, or it can result in deflected trajectories. The result depends on the speed of the colliding LMs: if it is greater than 0.29 m/s, then the two original LMs merge into a new one, otherwise they simply rebound. This value is therefore a lower bound for the speed of LMs. For this reason, in the logic gates based on LMs collision proposed so far in the literature [21, 22], three possible output channels have been considered when having both LMs as input.

Whether the LMs are merged or rebound from each other, the presence of at least one LM on one of these middle channels denotes the presence of an LM on both input channels. As a consequence, these possibilities are all logically equivalent. Since in the following large circuits will be treated, we will assume that LMs always have a speed greater than 0.29 m/s. This will always result in a new merged LM in case of a collision between two input LMs.

The design of the circuits we present in the rest of the paper will be explained through some schemes, defined as it follows. In order to keep the size of the schemes small and the reading complexity low, the above-mentioned interaction gate will be represented by means of a simple connection among input lines, since there will be a large number of such gates in the schemes (see Fig 1(a)). When both LMs ($A$, $B$) will be present in input, then the result of their collision will produce a LM on the center vertical line ($AB$) if only one LM is present in input, then the collision do not occur, and the LM will run across the opposite output line ($A\overline{B}$ or $\overline{A}B$).

Merging, in the case of both inputs present, is a prerogative of LMs travelling at speeds greater than 0.29 m/s. This phenomenon is not observable in the BBM, which consists of solid objects. Consequently, circuit design is simplified and requires significantly fewer circuit lines. Diagrams show that having diagonal inputs and a single vertical output is simpler compared to

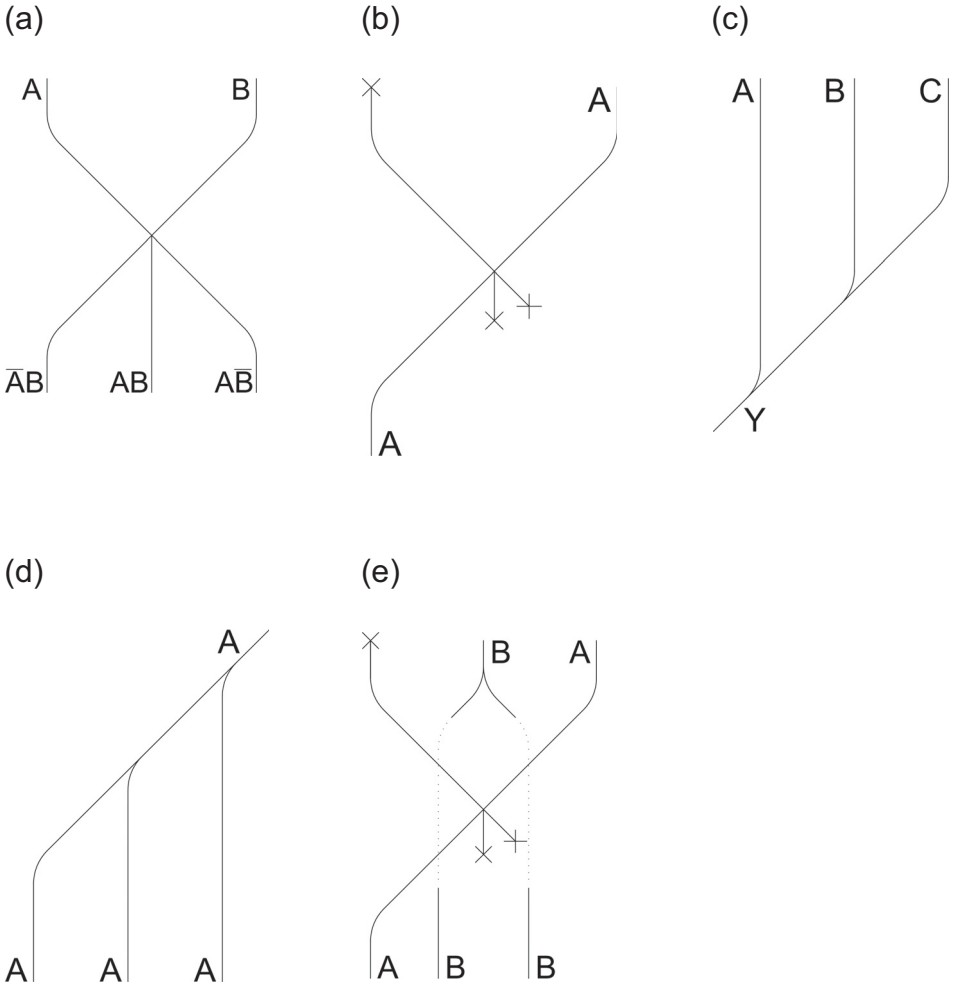

**Fig 1. Interaction gate (a), *EXIT* channels (b), union of multiple paths (c), routing of multiple paths (d), overlapping of multiple paths (e).**

implementations of the same circuits using BBM. In this last case, it is impossible to observe the slope of paths to distinguish between a single input and one that has been involved in a collision, since these are always vertical or horizontal. One can easily see the increase in simplicity and readability between, for instance, images Fig 8(a) and 8(b), which represent the same circuit.

Multiple speeds and coatings for LMs were analysed in the literature, allowing to highlights different behaviours. In [13], the focus is on coalescence and rebound phenomena during vertical collisions by varying the volume, impact velocity, and offset ratio of LMs. Meanwhile, in [24], the emphasis is on non-coalescent collisions at speeds between 0.12 m/s and 0.3 m/s, using a different coating of lycopodium and Teflon, which prevents LMs from merging.

In [14] one considers LMs travelling up to 1.11 m/s and colliding with a stationary LM. This is not exactly the case considered in this paper, but it shows how LMs can reach and interact at speeds much higher than 0.29 m/s. However, it is observed that at speeds greater than 0.6 m/s the LMs deformed into unusual shapes.

We thus will consider, in the following, 0.6 m/s as the upper speed limit, as the behavior of deformed LMs is not well studied and it is preferable to be avoided. A suitable general speed to consider is around 0.45 m/s, a reasonable compromise between 0.29 m/s, below which LMs do not merge, and 0.6 m/s, above which LMs deform.

Since, as explained above, LMs having a speed greater than the minimun value of 0.29 m/s is considered, a superhydrophobic scalpel would also be required to divide the LM generated by merging the two input LMs [19]. For the sake of simplicity, this is not represented in the scheme.

As in [22], we considered the use of an *EXIT* channel into which the discarded LMs are routed, so that there is no dispersion of LMs in the system, and they can be collected to be eventually reused. The presence of a path ending in an *X* represents this *EXIT* channel, as in Fig 1(b). Some lines will also start with the symbol *X*, representing an empty path that will never contain any LMs. An example is represented in Fig 8(b), where the reset channel *R* allows a null value, generated precisely by an *X*, to enter the first *D latch* to reset its value to zero.

For some circuits, different paths flow into the same one, but without corresponding to an interaction gate. In this case, we will use a representation like that in Fig 1(c): the two input paths do not collide with opposite directions, and that there are no 3 underlying outputs, typical of gates. The operation is straightforward: regardless of whether an LM is present in the first path or the second, it will always proceed to the single exit. These path joins are considered for situation in which it is structurally impossible for LMs to be present in both paths simultaneously, which would otherwise result in an unmanaged collision.

In Fig 1(d) there are paths that split, that is, where two or more lines emerge from one line. Making an analogy with classical electronic devices, it is nothing more than a node that divides the current input line into several wires. Just as this needs, in standard electronic gates, energy input from outside, in our model it requires to route copies of the LM present in input into all output paths. This can be done by splitting the LM with a hydrophobic scalpel [19] and replenishing its liquid content, or with a sensor [25] that reads the passage of LMs and communicates with a syringe designed to deliver them, as explained for the *SR Latch* in [22]. However, these solutions require an important amount of various resource; for this reason, circuits have been structured trying to avoid LM routing as much as possible.

In the schemes we will present, sometimes LM routes can also overlap. This will be represented by means of dashed lines, such as depicted in Fig 1(e). We want to stress the fact that, since we will be working with circuits in three dimensions, this does not result in a problem.

The datapath to be realized contains large circuits, often consisting of identical blocks repeated several times. For this reason, the diagrams we will considered in the following will

make use, where possible, of a limited number of inputs; the description will be then followed by an explanation of how it might be extended to the required amount of bit (which we will usually consider 32). Where this is not possible, the repetition of wires, components, or layers of a circuit in a sizable schematic is represented by three circles (see, for example, Fig 5(b)).

As usually done for classical boolean gate circuits, a channel with a diagonal line and an adjacent number indicates that it comprises multiple parallel channels rather than a single channel with an LM. The number beside the line specifies the quantity of channels.

For each circuit pattern, an approximate measure of the space it occupies will also be specified. The evaluation of the required space is based on the circuits presented in [26], where a full adder requires about 15cm x 15cm, and a single gate approximately requires 5cm in both height and width.

Regarding depth, this is not explicitly stated in the work just recalled. In [20], a thickness of 3mm is mentioned, which is sufficient to run LMs. The additional component that needs the most space is the LM-generating syringe, estimated to have a dimension of 3cm. This will therefore be taken as the required deep size of each layer. In the case of dashed paths (that is, overlapping paths), the estimated thickness of the circuits will not be increased. This is because the LMs occupy a negligible amount of space compared to the syringe, and it is therefore considered possible to place two of them side by side without increasing the thickness.

Once clarified the schemes used to represent the circuits, we are ready to propose a possible implementation of a multi-cycle datapath, like that presented in Fig 2, with all its components. Many of these components are internally formed by sub-circuits that are repeated several times, such as *multiplexers*, *decoders*, or *flip-flops*. We will therefore begin by analyzing these basic components in the next section.

## Combinational circuits

### Multiplexer

We first present the design of basic combinational circuits that will later be used for more complex circuits, starting by *multiplexers*. A *multiplexer* has n control lines ($S0$, $S1$, . . .,$Sn$), selecting which of the $m * 2n$ input channels ($A0$, $B0$, $A1$, $B1$, . . .) will be carried in the $m$ output channels ($Y0$, $Y1$, . . .), where $n$ is the number of inputs and $m$ is the length in bits of each input. In Fig 3(a) a 2 input *multiplexer* is depicted (n = 1, m = 1); Fig 4(a) represents a 4 input *multiplexer* (n = 2, m = 1), while Fig 4(b) represents a 2 input *multiplexer*, but such that each inputs has 2 bits (n = 1, m = 2).

The operation of the multiplexer in Fig 3(a) is straightforward: if an LM is not present in the control channel $S$, then the output channel $Y$ contains the value present in $A$. In fact, if an LM is present in $A$ this will go directly to the output $Y$. If, on the other hand, there is no LM present in $A$, then $Y$ will be empty. In all cases, any LM on channel $B$ would be lost on the discard channel. If, on the contrary, an LM is present in channel $S$, this will be carried to the second interaction gate, whether or not it collides (in the first interaction gate) with a possible LM present in $A$. If an LM is present in the channel $B$, a collision will occur thus bringing an LM in the output channel $Y$. If the LM in $B$ is not present, then the LM in $S$ will be removed from the circuit via the *EXIT* channel on the right. We note that, in this circuit as well as in the next two, there are different paths flowing into a single one. This does not cause any problems since, as explained in the previous Section (see Fig 1(c)), in all of these cases it is impossible by construction to have LMs on both paths at the same time.

A simple simulator based on the Billiard Ball Model (BBM) [23] was used to test this circuit. The same simulator has been used in [17] in an extended version, for the simulation of an *OR* gate and, subsequently, a *Toffoli* gate.

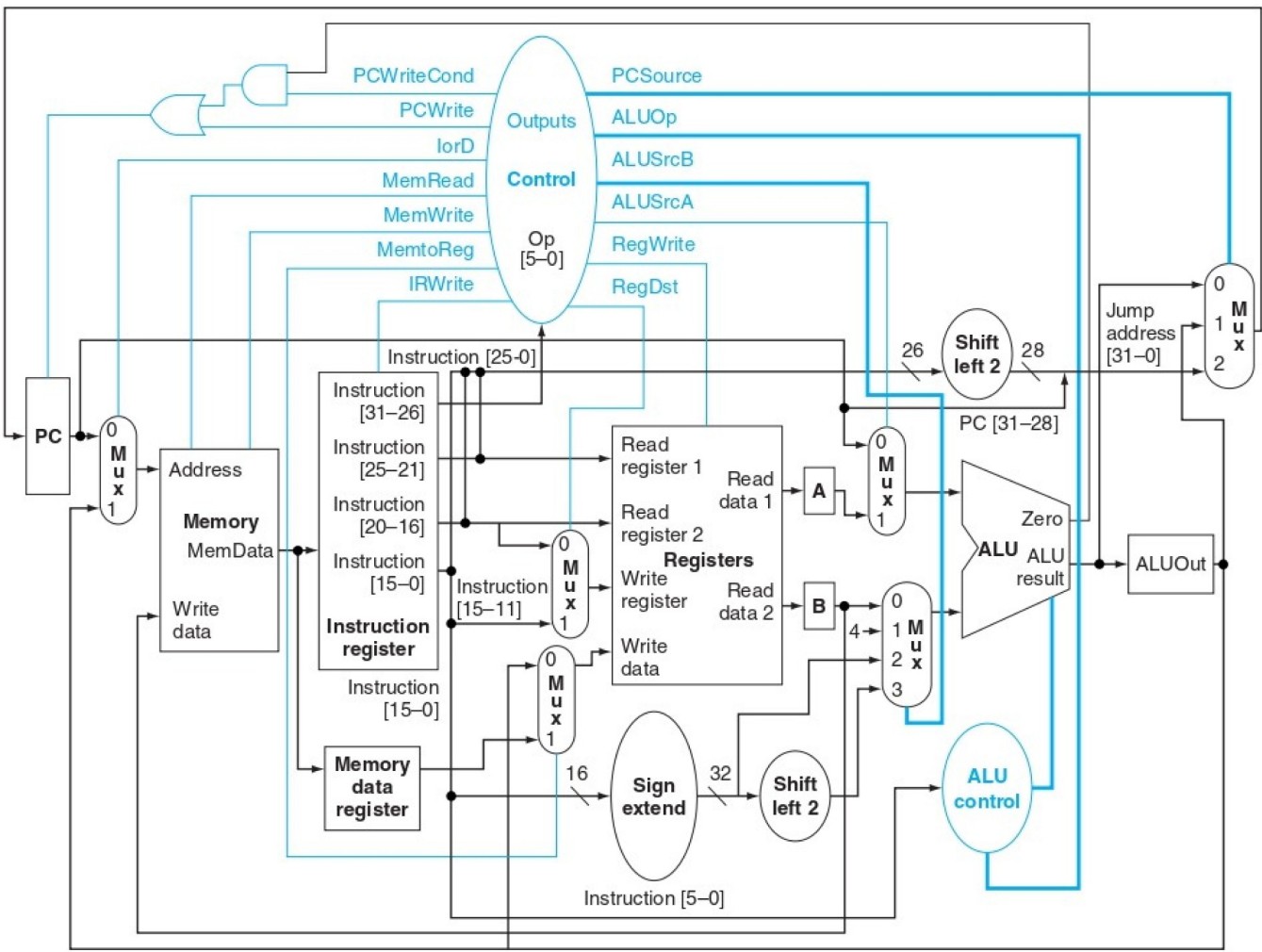

**Fig 2. Multi-cycle datapath schema** [27]**.**

In our case the BBM is a good comparison to the way LMs were used, glossing over the elastic and fluid-transport properties. Compared to BBs, the only substantial difference is the merging of two LMs, which have the advantages highlighted above. This simulator still ensures that the proposed schemes work on a theoretical level, having assessed their logical correctness, but it has a limited capacity and can only be used for circuits of modest size.

As already mentioned in the introduction and highlighted in [21], the main difference in the two models is the exit points of the colliding particles compared to the non-colliding particles. In BBM, the rebound is immediate and the $AB$ path is outside the $A\overline{B}$ and $\overline{A}B$ paths. With LMs, which follow the SSM, time is required for elastic rebounding, and the $AB$ path turns out to be inside the unchanged $A\overline{B}$ and $\overline{A}B$ paths. The whole problem can be ignored since we are always dealing with LMs that merge, falling vertically.

Fig 3(b) illustrates the functioning of a 2 input *multiplexer* presented in Fig 3(a). The red Billiard Balls (BBs) represent the inputs, while the white triangles represent the direction of their movement. The yellow BBs represent the channel in which a ball must be present to produce an output. BBs that leave the diagram are considered to be lost in the *EXIT* channels. As in the diagram in Fig 3(a), there are two *Y* channels, but there is no input configuration that

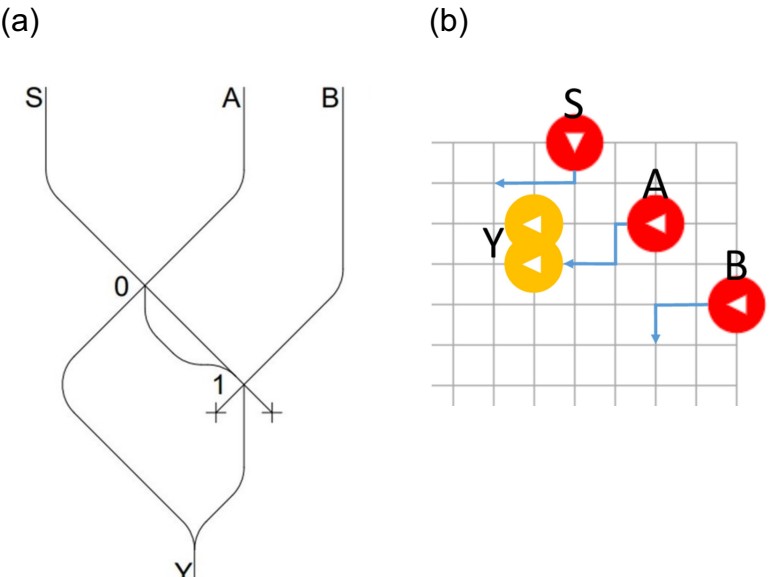

**Fig 3. 2 input *multiplexer* (a), schema BBM (b).**

generates a BB on both. In the diagram depicted in Fig 3(b), the movement of BBs is illustrated assuming the presence of all inputs (*S*, *A*, *B*).

In Fig 4(a), it's evident that each of the four inputs exhibits a different combination of channels in the gates to reach the output. The alternating 0 and 1 patterns repeat among them,

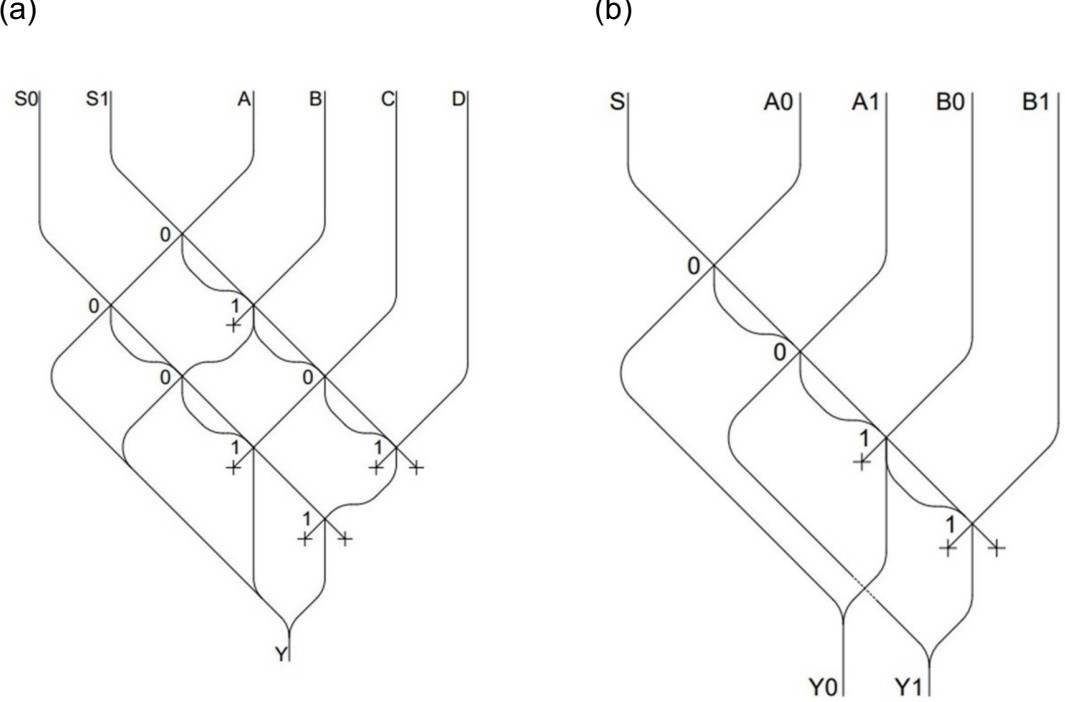

**Fig 4. 4 input *multiplexer* (a), 2 input of 2 bits *multiplexer* (b).**

generating all the four possible combinations achievable with two control signals. As the number of inputs increases, the method of collecting and routing LMs to the underlying plane will remain consistent. This pattern of 0s and 1s will be extended to 32 inputs and 5 control channels. A further step is needed to effectively store and manage data: in fact, each of these inputs must consist not only of a single bit, but of 32 bits. An illustration of how this can be accomplished is presented next.

As illustrated in Fig 3(a), in Fig 4(b), we still have two input channels, but consisting now of two bits each. The control signal *S* remains a single one, and the pattern for all bits of the same input remains the same, with the difference being the output channel to which they are carried.

Since we planned the design of 32 bits based circuits, we need a *multiplexer* with 32 inputs of 32 bits each, and thus requiring 5 control inputs. This would then need 1024 gates, successively placed one after the other; the approximate size of one such a circuit would be about fifty square meters. Apart from the obvious space problem, it is crucial to emphasize that a taller device implies longer computation times. Typically, inputs are positioned at the top of the circuit, and outputs at the bottom, and a computation requires LMs to traverse the circuit entirely. A possible solution to this huge size and, as a consequence, to the computing time is thus proposed in the following.

## Stratification and stratified multiplexer

Considering both the computing time and the huge dimensions of the *multiplexer* (consisting of 32 inputs channels, having 32 bits each) required to design a multi-cycle datapath according to the circuit described in the previous section, an alternative circuit design method is proposed. The idea behind this different approach is to divide the circuit into many smaller ones, and place them side by side. This results in a kind of LM beehive where sub-circuits are active in parallel, allowing the break down of the height dimensions, moving them into depth, while parallel computations also allow improvements in computation time. One can use a routing device similar to the one presented in Fig 5(a), that runs on different paths for LMs put in parallel with each other. In this case, starting from one path three new paths are obtained, each present in a different circuit. LMs *B*, *C*, and *D* are blocked by the electromagnetic mechanism as explained in [21], waiting to be allowed to proceed. Path *A* is then chosen as the main one: with the arrival of an LM on this path Fig 5(a.1), its weight will result in the rotation of the device Fig 5(a.2) and the subsequent fall of all LMs Fig 5(a.3). This solution requires an LM generator on each path, and the proposed schemes would change drastically.

With this technique it is possible to implement two different 32 input *multiplexers* of 32 bits each. In the first there are 32 layers, one for each output bit, where each layer is handled exactly like a 32 input of 1 bit *multiplexer*, similar to those shown above. Placed all side by side they form 32 output bits. The 5 control signals use the routing device mentioned above in order to split into all 32 circuits.

Another possibility is to make the circuit as simple as possible, thus handling only 1 input bit, and copying each circuit configuration 32 times, one for each bit. In Fig 5(b) it was shown how the first bit of the first input is handled, which is to be active with control configuration 00000, and every other bit of this input will be handled equally (there are 32 of these). The circuit will change for each layer of the second input, this being active with a control configuration of 00001. This configuration makes the circuit small, but obviously requires a large number of aligned circuits, 1024 to be exact, as well as a very long routing device that can cover them all. It could be particularly advantageous if there is the

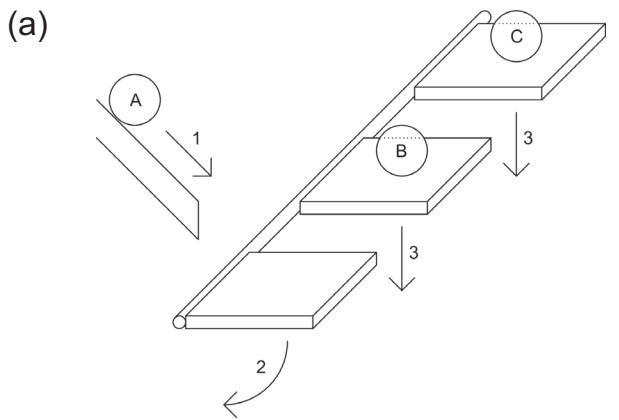

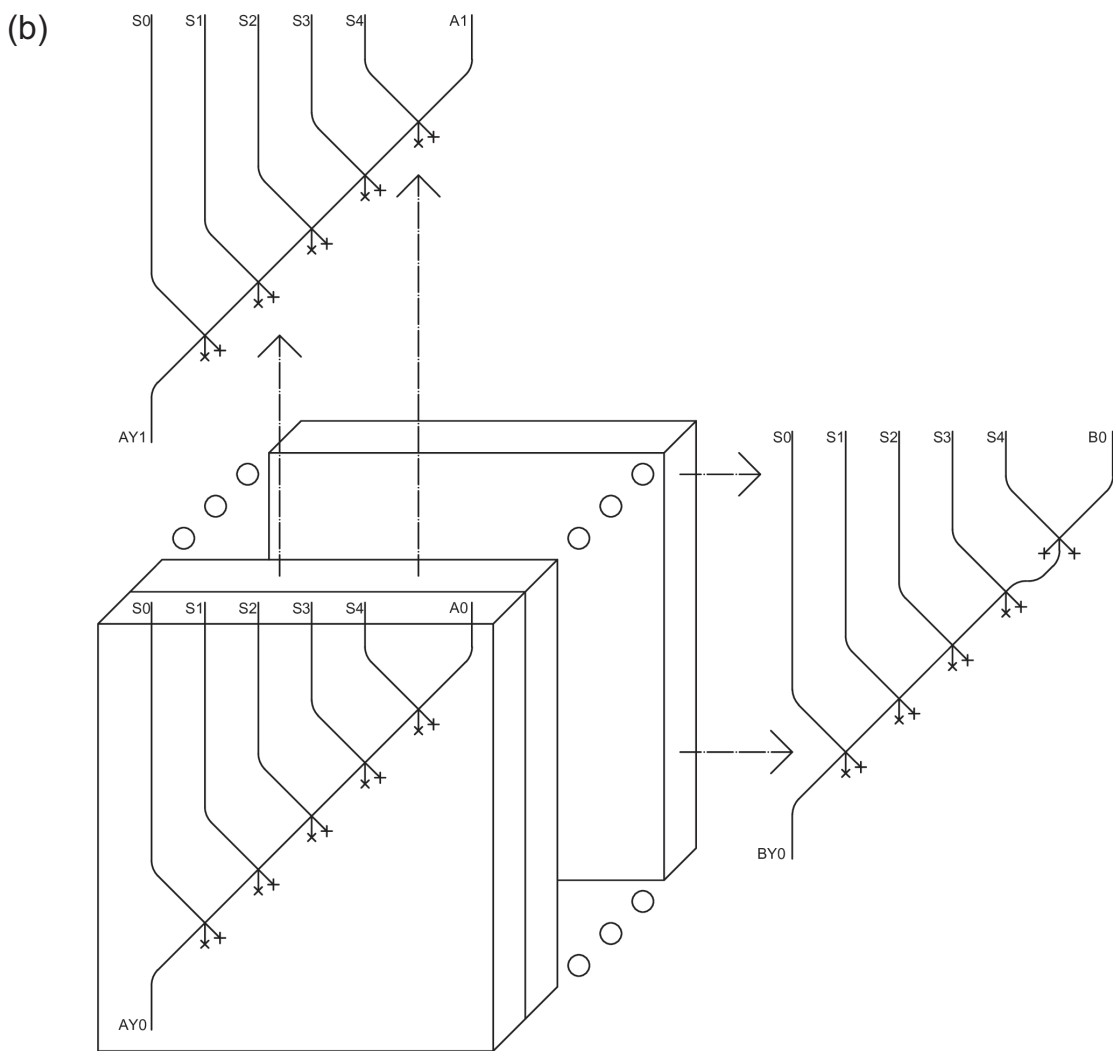

**Fig 5. Routing device (a), stratified *multiplexer* (b).**

possibility of mass production of small circuits and an inexpensive way to create LM generators.

An additional detail to keep in mind is that, at the design stage, a decision should be made whether to follow a single-layer, 32 layers or 1024 layers implementation in order to achieve maximum benefits while maintaining the same structure. Considering 3cm for each layer, a 32 layer implementation will have about 1 meter depth, while a 1024 layer implementation will have about 30 metres depth. Using this method for implementing a 32 input *multiplexer* of 32 bits each, the size of 1.8m x 1.8m x 1m on 32 layers, or 0.25m x 0.25m x 30m on 1024 layers, can be achieved.

Of course stratification, despite the advantages just described, also entails certain problems that cannot be ignored, and that do not make it an obvious implementation solution. In particular, in stratified circuits, the property of reuse and minimisation of LMs is lost, as each small circuit is independent and replicated. Moreover, for circuits such as the *ALU* that will be presented in Section, it is not possible to adopt this solution efficiently since, as we will see, each block needs information produced by the previous block in order to correctly execute computations. We will further highlight such issues in the following sections, when discussing each specific circuit.

## Decoder

The *decoder* is another combinational device that will later be used in various schemes. This device requires an LM-generating syringe, synchronized with the rest of the circuit, as showed for the *NOT* gate in [22]. The solution of exploiting a syringe that generates LMs unconditionally will also be used in sequential circuits (Figs 8 and 9) and in the *ALU* (Fig 13). The schemes shown in Fig 6 are the ones, among the many analyzed, that minimize the use of interaction gates. Only the scheme of a *decoder* with a small number of inputs is presented. Although larger circuits are needed, a sort of recursive definition can be derived: by adding an input signal to the *decoder* with *n* inputs and crossing it with all the outputs, a *decoder* with *n+1* inputs can be obtained.

For instance, the 3-input/8-output *decoder* in Fig 6(b) has been obtained in this way, starting from the 2-input/4-output *decoder* in Fig 6(a).

In general, a *decoder* with n inputs and $2^n$ outputs requires $2^n$-1 interaction gates. For the definition of our datapath, a 5-input decoder is needed for the *register file*, and a 6-input one for handling the OPcode in the *control logic*. As a consequence, the resulting circuits will have approximate sizes of 0.8 and 1.6 square meters, respectively, each with a depth of 0.03m. Note that the waste paths represented by *X* will never contain an LM, by construction.

## Stratified decoder

As already done for multiplexer, stratification can be effectively used to lower the height of decoder circuits. An implementation of the 5-input *decoder*, stratified into 32 layers, is proposed in Fig 7. In each layer, a specific sequence of 5 inputs is analyzed, and an LM is produced in output if and only if that precise sequence is present. The input case 00000 involves a special configuration, specifically requiring a syringe generating LMs, since no LMs are present in any input channel. Even though the size of a non-stratified 5-input *decoder* is not as large as that of other components, using a stratified design might still be advantageous to maintain a consistent number of layers throughout the entire datapath. The downside of this approach is the significant number of LM splittings required. Using the 32 stratified layers circuit design, a circuit of size 0.25m x 0.25m x 1m is achieved.

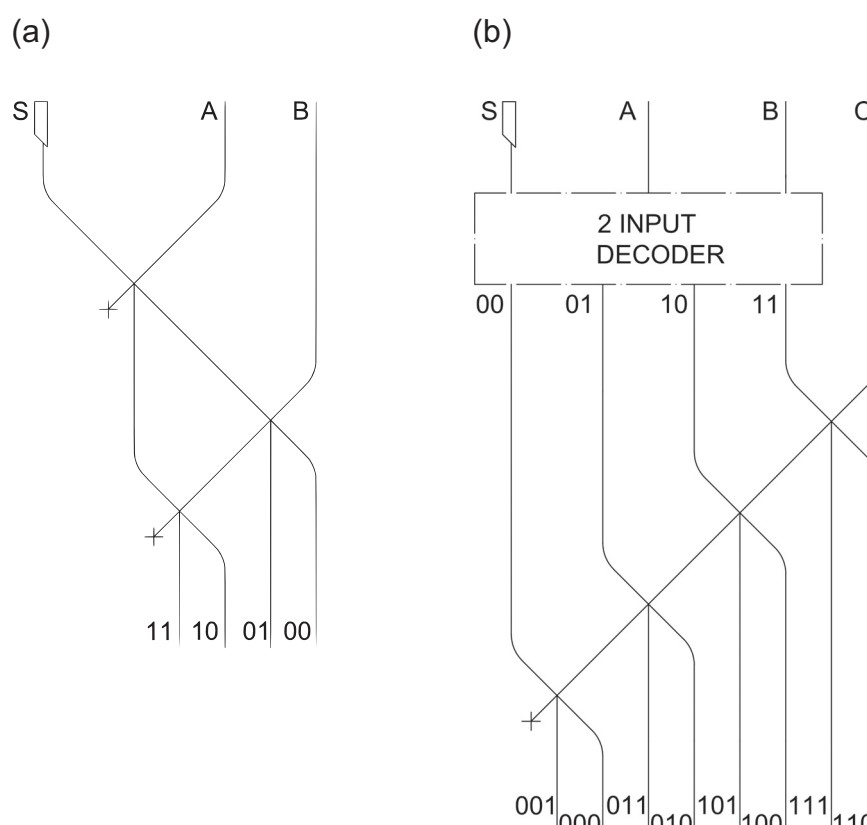

**Fig 6. 2 input *decoder* (a), 3 input *decoder* (b).**

## Sequential circuits

### D latch, D flip-flop

This section discusses sequential circuits, which allow data to be saved in memory. We will start with the *D latch* (depicted in Fig 8(a)), then the *D flip-flop* (Fig 8(b)), and the *register files* (Fig 8(c)), analyzing their read and write handling.

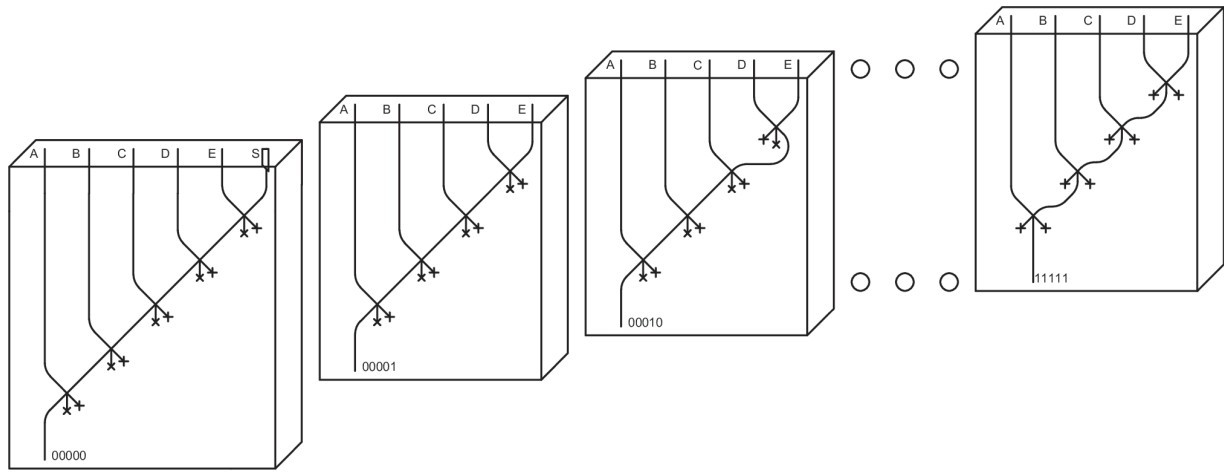

**Fig 7. 5 input *decoder* stratified on 32 levels.**

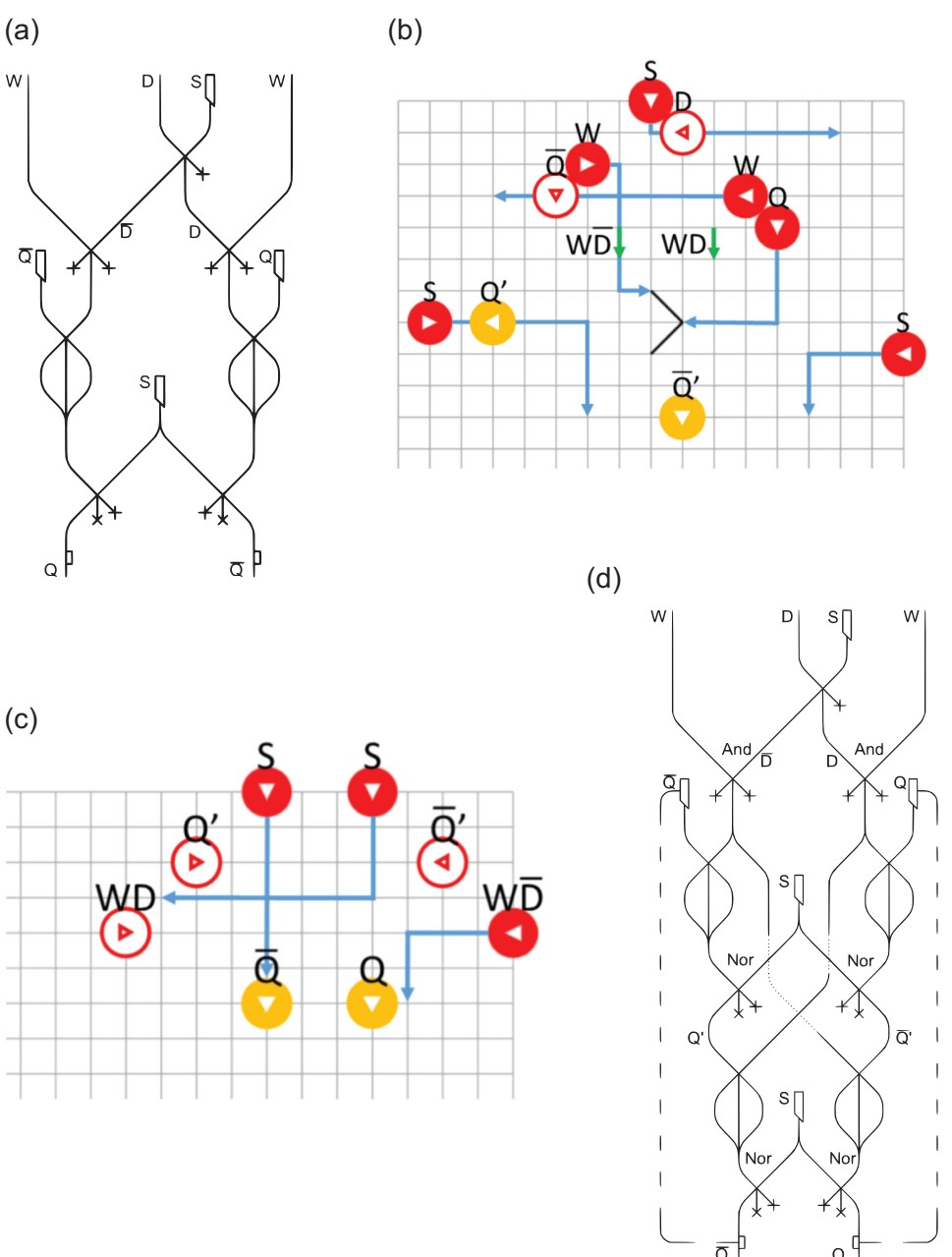

**Fig 8. Direct implementation of *D latch* (a) BBM *D latch* (b) BBM additional module (c) correct implementation of *D latch* (d).**

In classical electronic circuits these types of schemes require the introduction of a square-wave signal, the so called clock, which regulates the synchronization between input and output. In our case, a clock signal is already considered, since it is required to coordinate all the interaction gates in the datapath. In fact, in all the schemes, and not only those in which the clock is strictly required, the LMs will have to be released with the right timing to avoid non-collision due to incorrect synchronization. This particular topic is not discussed further in this paper. There remains, however, a need for a write signal *W* that enables the write in the *D latch*, and that allows the device to keep the data in memory in case the *W* value is absent.

The D latch is of particular interest because it highlights the differences between classical electronic circuits and those presented in this work. A classical D latch has outputs connected to inputs, allowing the signal in the circuit to stabilize through this feedback mechanism. By directly translating the circuit with interaction gates and LMs, this feedback is no longer immediate. Incorrect LMs, representing a temporarily unstable signal, are brought into the output, resulting in erroneous outcomes. To identify and study this error, the use of the BBM simulator [23] was crucial.

Initially the design of an *SR latch* [22] was considered and adapted to design a *D latch*. The result, shown in Fig 8(a), can be obtained by changing the inputs to $D$ and $\overline{D}$ instead of $S$ and $R$. A *NOT* port [22] performs this operation on $D$; the resulting channels go through an *AND* gate with the write input $W$, and these are then processed by a *NOR* gate with the $Q$ and $\overline{Q}$ signals stored in the device.

This trivial translation leads to the error anticipated above. For instance, when setting a value of $Q$ that is not present in the device memory ($W$=1, $D$=1, $Q$=0) or resetting one that is present ($W$=1, $D$=0, $Q$=1), the output signals $Q$ and $\overline{Q}$ would both be null. A demonstration of this, with inputs $W$=1, $D$=0, $Q$=1, is presented in Fig 8(b). This configuration of BBs represents the pattern of the *D latch* in Fig 8(a). The white BBs represent the positions that any inputs, currently not present, would occupy.

This problem also occurs in classical circuits but is immediately resolved by the fact that the inputs to the two *NOR* gates are updated from the outputs. When dealing with LMs, this is only possible if a second control is performed schematically.

To address this issue, it is therefore necessary to add the block depicted in Fig 8(c) after the one in Fig 8(b). The input BBs $WD$ and $W\overline{D}$ are derived from the channels highlighted in green in Fig 8(b). This configuration ensures the accurate implementation and sustained functionality across all feasible inputs. The diagram in Fig 8(d) has been implemented accordingly. A secondary traversal through the ports *NOR* has been effectuated using $Q'$ and $WD$ as inputs for the former, and $\overline{Q'}$ along with $W\overline{D}$ for the latter, as previously shown in the simulator.

In the case of a *D latch*, a further complication also arises: an output must be connected to an input. While this is easily done in classical electronic circuits, dealing with LMs would require making an LM go up in a circuit, going against the force of gravity.

This could be solved with mechanical solutions, such as an LMs elevators or links between input and output. A method of handling LMs can be found in [20] which, although it serves to keep track of the passage of an LM in the path, shows how devices based on mechanical movements can be amalgamated in circuits, achieving excellent results. At present, there is no such device in the literature, so its implementation is left as an open question. In the rest of this work, this will be generically implemented by means of a sensor connected to a syringe that generates LMs. If the sensor detects an LM passing through, then it will activate the syringe that will generate one new LM, simulating the LM being re-inserted to the top channel.

The sensors are denoted in the figures by means of small rectangles near the output channels, which store information about the passage of LMs in $Q$ or $\overline{Q}$, and generate one new LM in the corresponding syringe.

Each syringe is labelled with the same letter as the sensor to which it is connected, and can be seen from the dotted lines in Fig 8(d).

The union of two *D latches* generates a *D flip-flop* (Fig 9(a)), to which a *multiplexer* has been added with a reset $R$ value for exception and error handling. Each *D flip-flop* keeps only one value, i.e., only one bit, in memory. For the purpose of creating a multi-cycle datapath, however, this is not sufficient. Placing several *D flip-flops* side by side as in Fig 9(b) allows the creation of a memory cell, in this case a 4 bit *register*, used as an example. As usually,

(a)

(b)

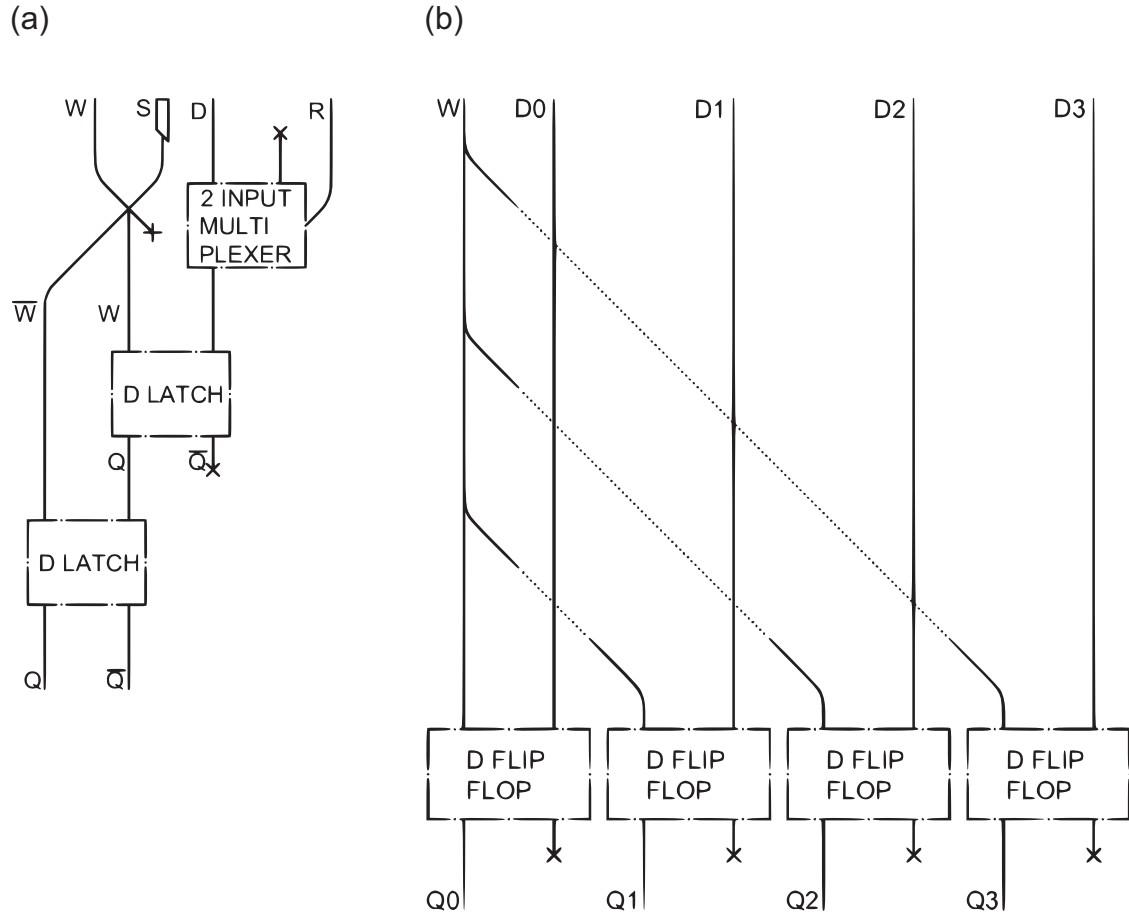

**Fig 9.** *D flip-flop* (**a**) **4 bit** *register* (**b**).

considering that data in our circuits consists of 32 bits, it is necessary to design memory registers of this size. It will thus be necessary to use 32 *D flip-flops* to create a multi-cycle datapath memory cell. The write signal *W* is routed into each of these. The dimensions of the resulting device would be of approximately 0.6m x 6.5m x 0.03m.

## Register file

A single 32-bit memory cell, however, is still not sufficient. The *register file* ("Register" in Fig 2), which serves as the basic memory for the multi-cycle datapath, contains 32 *registers*, each 32 bits in size. To construct this, a 32-bit *register* is created and replicated 32 times, each *register* having its own write *W* value. Consequently, the *register file* consists of 1024 bits (32 x 32) of input and output data, along with 32 bits for *Write*, one for each *register*. The overall dimensions of the *register file* are thus approximately 0.6m x 200m.

A width of 200 meters is, of course, impractical, necessitating the use of layering as a solution, as previously done. When creating a 32-bit memory cell, it has been explained how the 32 *D flip-flops* should be positioned side by side, with the write signal *W* routed to each one. Layering offers again an excellent solution, allowing for a circuit of a same size as for a single *D flip-flop*, but with a depth equivalent to 32 *D flip-flops*. This method also addresses the routing

of the 32 *Write* signals through the horizontal routing mechanism described earlier and depicted in Fig 5(a).

Merging these circuits will result in the definition of a 32 bit *register file*, where one could eventually decide to further proceed in layering, and thus scale up to 1024 layers. The resulting circuit would then have a size of 0.6m x 0.2m x 30m. If, on the other hand, we opt for an intermediate route, in each layer we would have a 32 bit memory cell, and by placing 32 of them side by side we get the 32 bit *register file*. The whole circuit have a size of 0.6m x 6.5m x 1m.

### Reading, writing and stratification of a register file

The technique for writing (Fig 10(a)) and reading (Fig 10(b)) data into the *register file* is now demonstrated using an example with four 4-bit inputs. To manage the *register file*, we use three 5-bit inputs (*RN*, *R1*, and *R2*) for addressing the *registers*, a 32-bit input (*RD*) containing a potential value to be written, the previously discussed Write (*W*) input, and two 32-bit outputs that will hold the values from the *registers* addressed by *R1* and *R2*. *RN* is used to select the address where the value in *RD* will be written.

The *RN* register number enters a 5 input *decoder* (2 in Fig 10(a)), so that the 32 possible outputs are obtained, and goes into *AND* with *Write* to check whether writing is possible. The LM

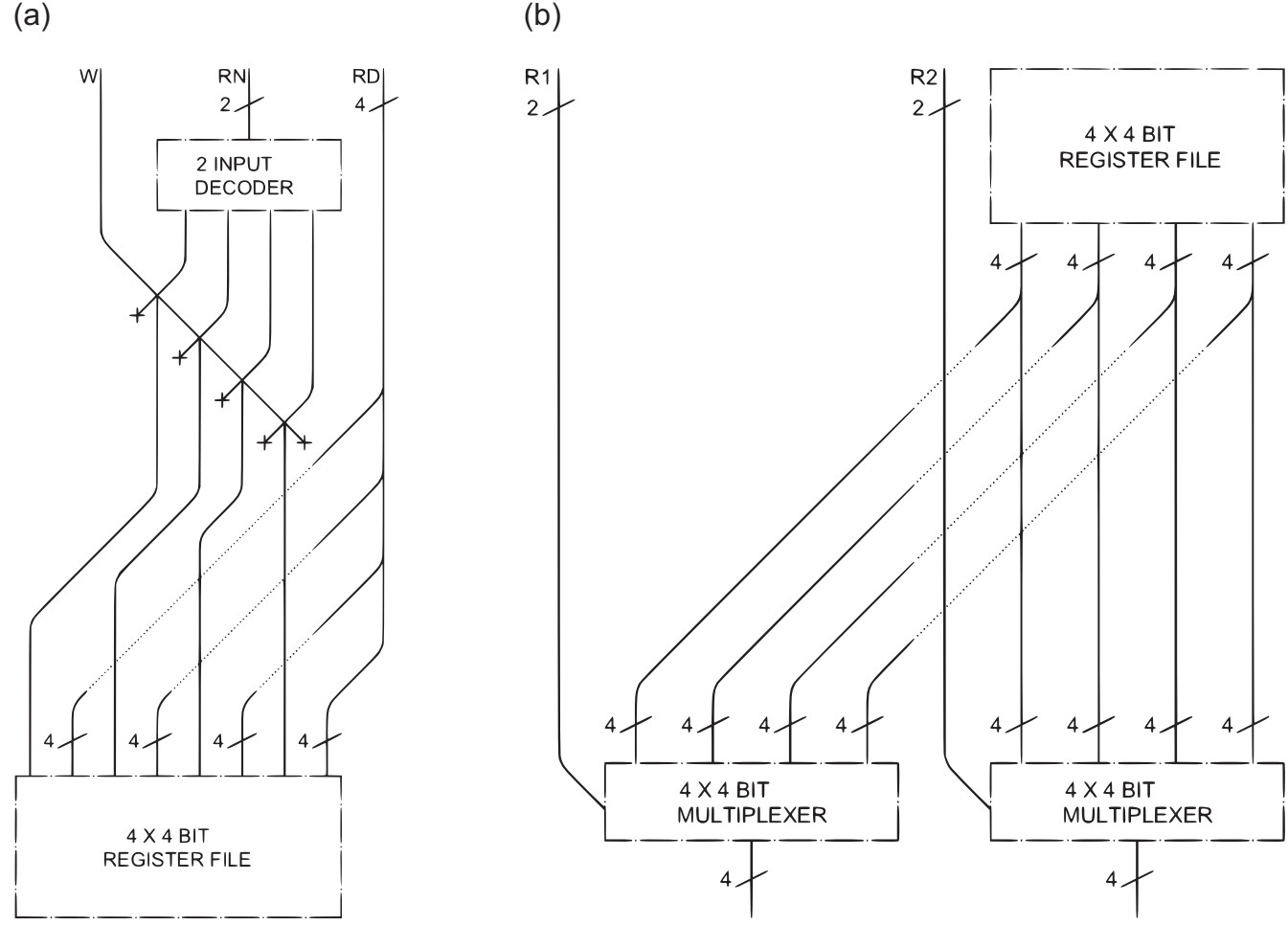

**Fig 10. Writing data into the *register file* (a), reading data from the *register file* (b).**

will then enter the *W* bit of the corresponding *register* and no others, ensuring that no further registers will be modified.

In order to read two of the values in the *register file*, two input addresses containing the location of the data to be read are required. For this task, 5 bits *R1* and *R2* (2 as simplified in Fig 10(b)) are used in order to reach all 32 memory cells of the *register file*. Each of the 32 outputs (4 in Fig 10(b)) of the *register file* will enter two multiplexers with 32 inputs of 32 bits each (4 inputs of 4 bits in Fig 10(b)) and output the required data. Altogether, the *register file* reaches an approximate size of 50m x 200m.

As previously mentioned, layering the *register file* offers some advantages, which are not diminished by the need of incorporating the *decoder* for writing and the *multiplexers* for reading data. This approach results in circuits of approximate dimensions of 4m x 6.5m x 1m with 32 layers, or 2m x 0.5m x 30m with 1024 layers.

## Memory and registers

The circuit memory (denoted as "Memory" in Fig 2) is the component of the multi-cycle datapath designated to store the instructions of the code to be executed. From here, instruction values are transferred to the *instruction register*, which then decodes and processes them as needed. A similar approach to the *register file* was employed to create the schema shown in Fig 11. The write phase operates in the same manner, but only one *multiplexer* is required for reading since only one instruction is read at a time. The three addresses (*R1*, *R2*, and *RN*) have been joined into a single *Address*. The read and write addresses are identical, with *MemWrite* indicating whether a line of code is being written or read. A second *multiplexer* uses *MemRead* as a control signal: if there is a load memory (LM) operation, then it will read the selected data; otherwise, it will output 32 null values derived from the *X*.

The "Instruction register" in Fig 2, is simply a 32-bit *register* that splits its output to facilitate a clearer depiction in subsequent steps, where these outputs will be manipulated. Consequently, it shares the same characteristics, problems, and size as a standard 32-bit *register*.

## ALU

### 1 bit ALU

We are now ready to present the schemes for a 32-bit *ALU*, which performs operations on 32-bit input and output data. To this goal, we first introduce a 1-bit *ALU* in Fig 12(a). Following our established approach, we place 32 of these 1-bit *ALUs* side by side, with a few minor adjustments. Specifically, 31 of these units will be identical, while the last one, *ALU31* in Fig 12(c), will include an overflow control as shown in Fig 12(b) and it will contain an additional output, denoted by *Sum*.

In order to handle the operations in the *ALU*, two control signals are needed to invert the values of *A* and *B*, called *Ainv* and *Binv*, respectively. If these channels contain an LM, then the corresponding value of *A* and *B* will have to be inverted. This can be schematized as for the *XOR* logic gate presented in [22].

The outputs obtained in this manner will have to be routed to a *multiplexer*, after combining them using *OR* and *AND* operations, and to a *full adder*, as proposed in [26]. In this way, the *ALU*'s internal *multiplexer* will be able to choose, based on the *Op* bits, which operation bring to the output.

In particular, the inputs of the *ALU* include the *A* and *B* signals, the *Ainv* and *Binv* controls that invert them, the *Cin* carryover from the previous *ALU*, the *Less* signal, which is almost always set to 0, and the two *Op* bits that specify the operation to be performed by the *multiplexer*. The pattern of operations is detailed in Table 1. The output will be the result of the

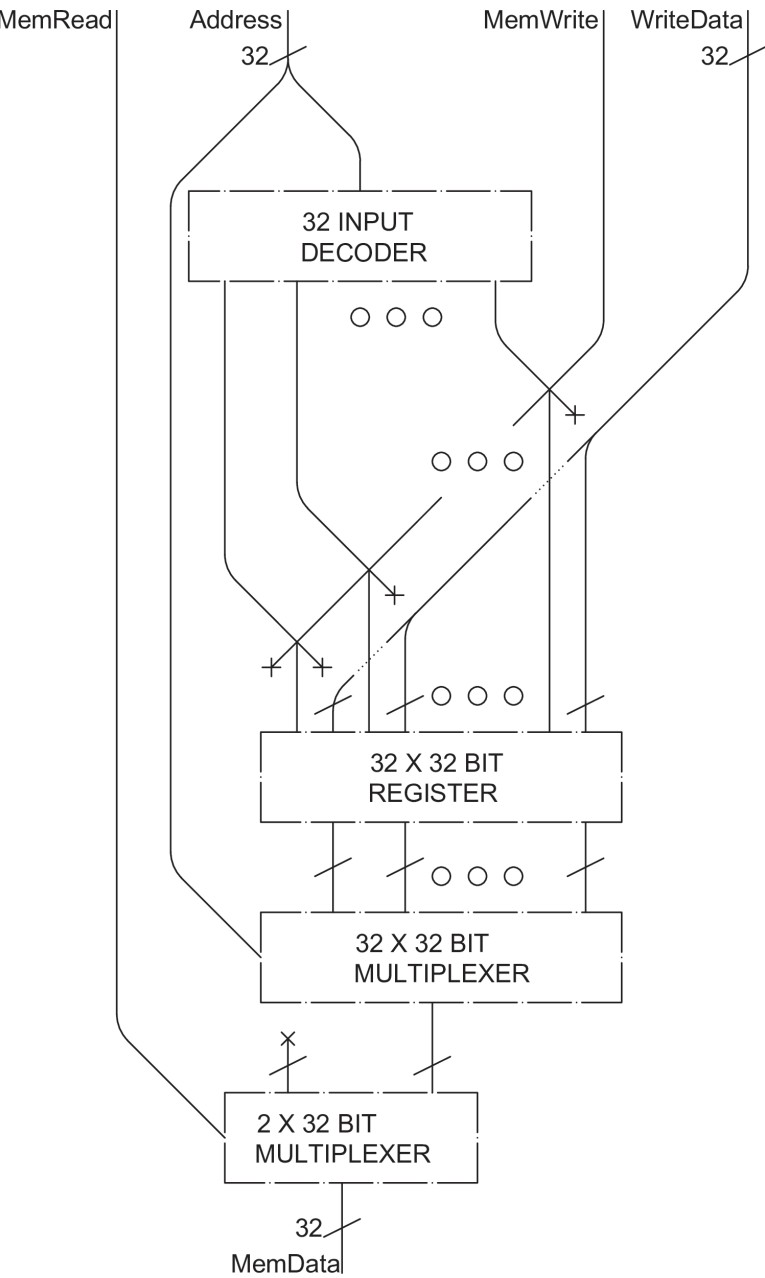

**Fig 11. Instruction memory.**

specified operation, along with any carryover from the sum (or subtraction, if *B* is negated). Since it consists of a small *multiplexer* and few logic gates, the stratified option is not considered for the single *ALU*. The resulting device has an approximate dimension of 0.5m x 0.5m x 0.03m.

## Overflow detection

As previously explained, the last 1-bit *ALU* requires an additional component, i.e. an *overflow detector* (Fig 12(c)). This device notifies when the result of sum or subtraction operations

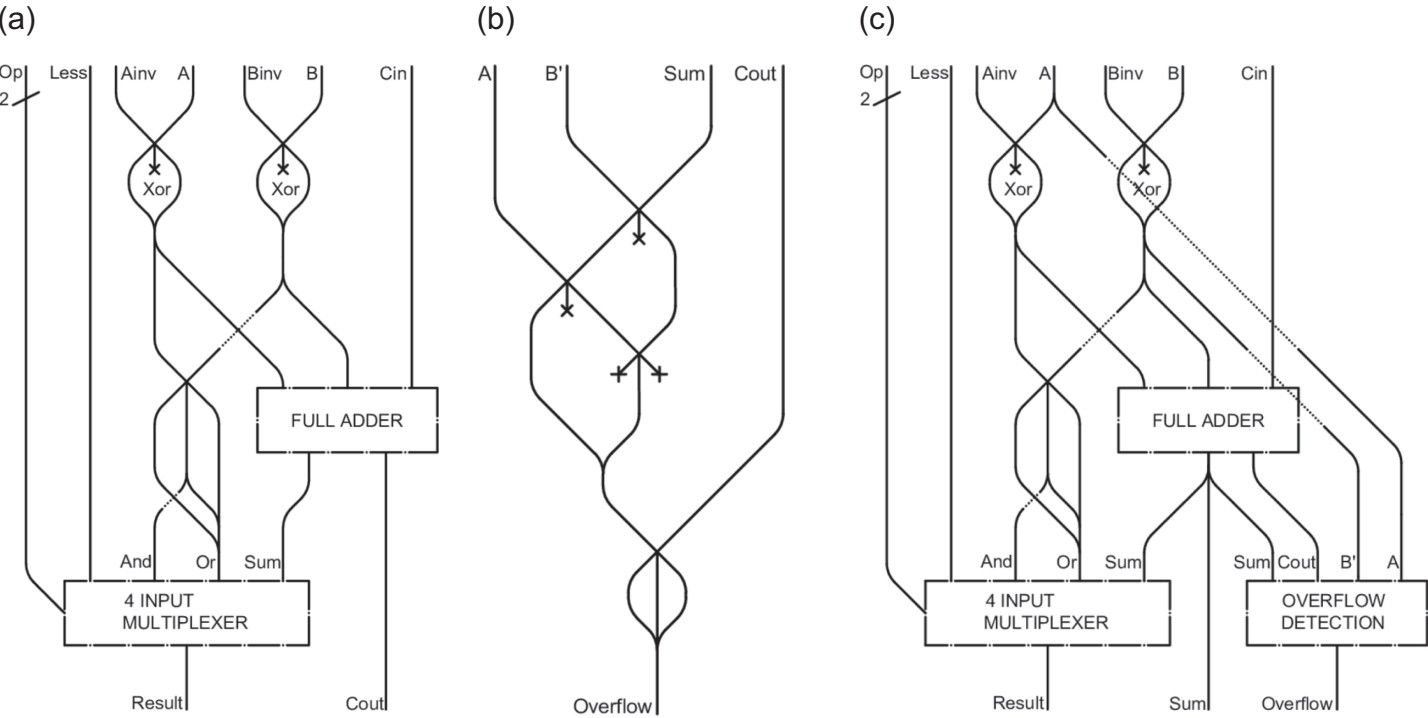

**Fig 12.** *ALU from 0 to 30 (a), overflow detector (b), ALU31 (c).*

exceeds the allowed limit. The circuit in Fig 12(b) is added to the standard 1-bit *ALU* and takes *A*, *B′*, *Sum*, and *Cout* as inputs to check for overflow in *ALU31*. *B′* is the value of *B* after a possible inversion by *Binv*. The overflow detection uses the Eq (1).

$$\text{Overflow} = ((A \wedge B' \wedge \overline{\text{Sum}}) \vee (\overline{A} \wedge \overline{B'} \wedge \text{Sum})) \vee \text{Cout} \tag{1}$$

If the most significant bits of *A* and *B′* are equal to each other, but different from the result bit *Sum*, then an overflow has occurred. The same outcome is obtained if, trivially, *ALU31* has nonzero value of *Cout*, that is, a remainder from the sum.

## 32 bit ALU

The next step involves merging the 31 identical *ALUs* with the thirty-second one featuring overflow detection, thus forming the complete 32-bit *ALU* as shown in Fig 13. This component has inputs *A* and *B*, both 32 bits in size, along with control signals *Ai* and *Bi* to invert their values, and the two *Op* bits to select the operation to be performed in the *ALUs*.

**Table 1. Different operations and respective code.**

| Op value | operation |
|---|---|
| 00 | AND |
| 01 | OR |
| 10 | SUM |
| 11 | LESS |

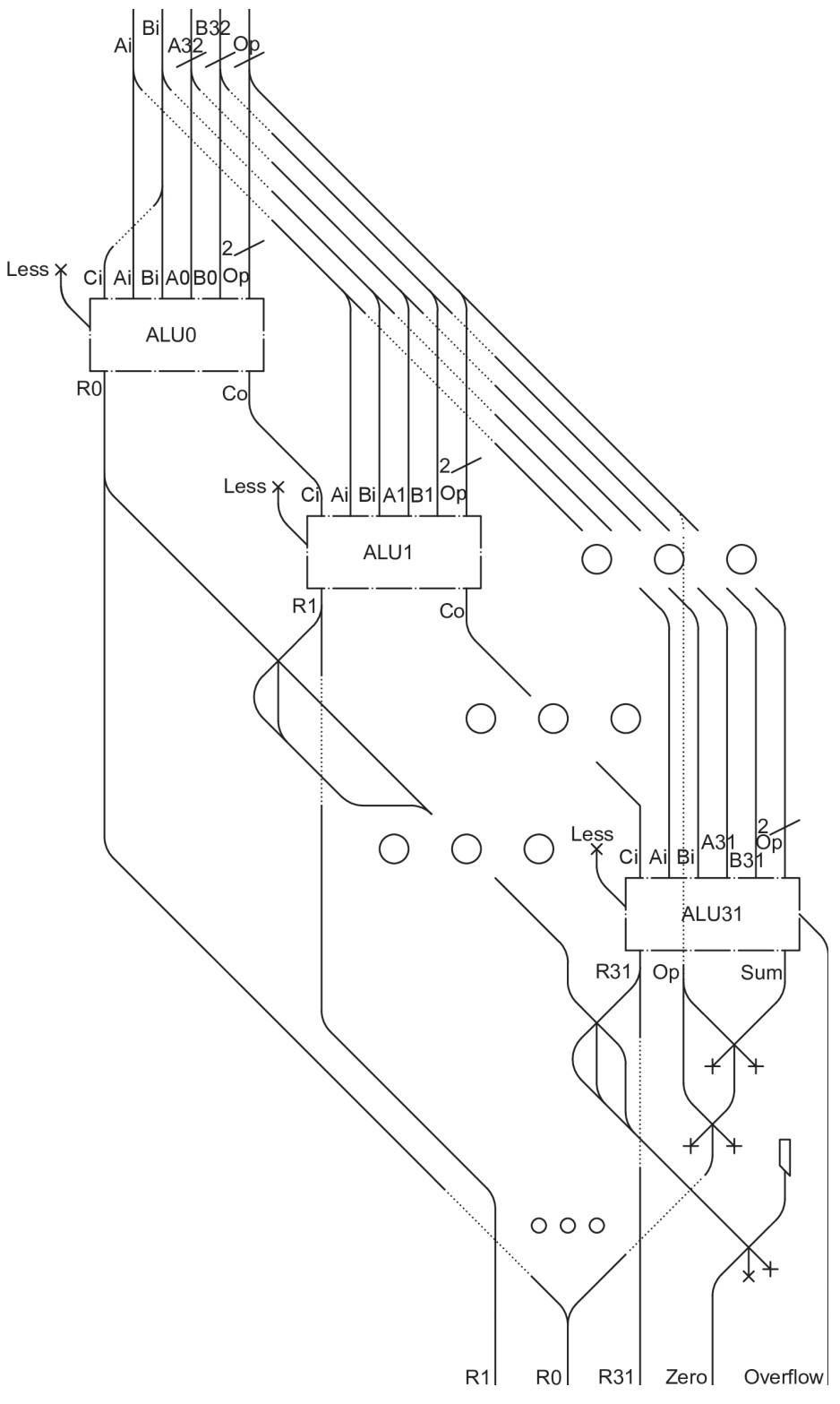

**Fig 13. 32 bit *ALU*.**

The *Cout0* of *ALU0* will become the *Cin1* of *ALU1*, and the result *R0* will have to be split into two paths: one that will go to represent the first output bit, while the other will have to collide with the results of all the other *ALUs* to check whether the 32 bit *ALU* gave a completely null result. This information is useful in the case of BEQ (Branch On Equal) operation, i.e., jumping in the case of identical *A* and *B* registers. This is done by checking if the difference between the two values *A* and *B* gives a null result. The 32 bit *ALU* therefore has 3 outputs: one set of 32 bits, containing the result, a *Zero* signal, which will contain an LM if and only if the *ALU* gave a null result, and an *Overflow* output.

The *Cout0* of *ALU0* will become the *Cin1* input for *ALU1*, ensuring the carryover from one *ALU* to the next. The result *R0* will have to be split into two paths: one representing the first output bit, while the other collides with the results from all other *ALUs* to check if the 32-bit *ALU* produced a completely null result. This information is useful fin the case of BEQ (Branch On Equal) operation, where a jump occurs if the values in *A* and *B* registers are identical. This is done by checking if the difference between the two values *A* and *B* gives a null result. Consequently, the 32-bit *ALU* has three outputs: one set of 32 bits containing the result, a *Zero* signal, which will contain an LM if and only if the *ALU* gave a null result, and an *Overflow* output.

We finally point out that, for *ALU0*, the inputs *Bi* and *Ci* are connected to the same path. This arrangement is necessary because, in the case of a subtraction operation, both inputs must be set to 1, whereas during addition and logical operations, they must be set to 0.

The total approximate dimension for the complete *ALU* circuit is 16m x 7m x 0.03m.

## Control logic

The last proposed component for implementation is the *control logic* circuit, responsible for analyzing the current instruction and transmitting suitable signals to execute it accurately. This control logic routes all controls depicted in Fig 2 of the multi-cycle datapath.

In order to correctly execute the process, the first 6 bits of the current line of code are required, denoted as *OP0*, *OP1*, *OP2*, *OP3*, *OP4*, *OP5*. Additionally, information on its previous state is needed, since each operation lasts three to five computing cycles. In each of these cycles the LMs will traverse the entire datapath, using the components proposed so far. The state is stored by the 4 channels *S0*, *S1*, *S2*, *S3*; a 4-bit *decoder* is employed to generate the necessary control outputs based on the current state. The outputs *S0*, *S1*, *S2*, *S3* represent the exact values that will be used as inputs in the subsequent step.

All values used in the next circuits are taken from the state diagram in Fig 14, specific to the Jump operation. Of course, different operations are associated with different states, each with their respective outputs. Additional information regarding possible operations and their codes can be found in [28].

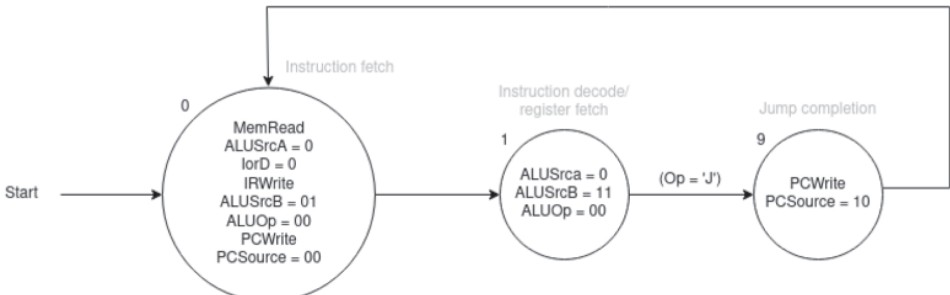

**Fig 14. Control logic state diagram for Jump operation.**

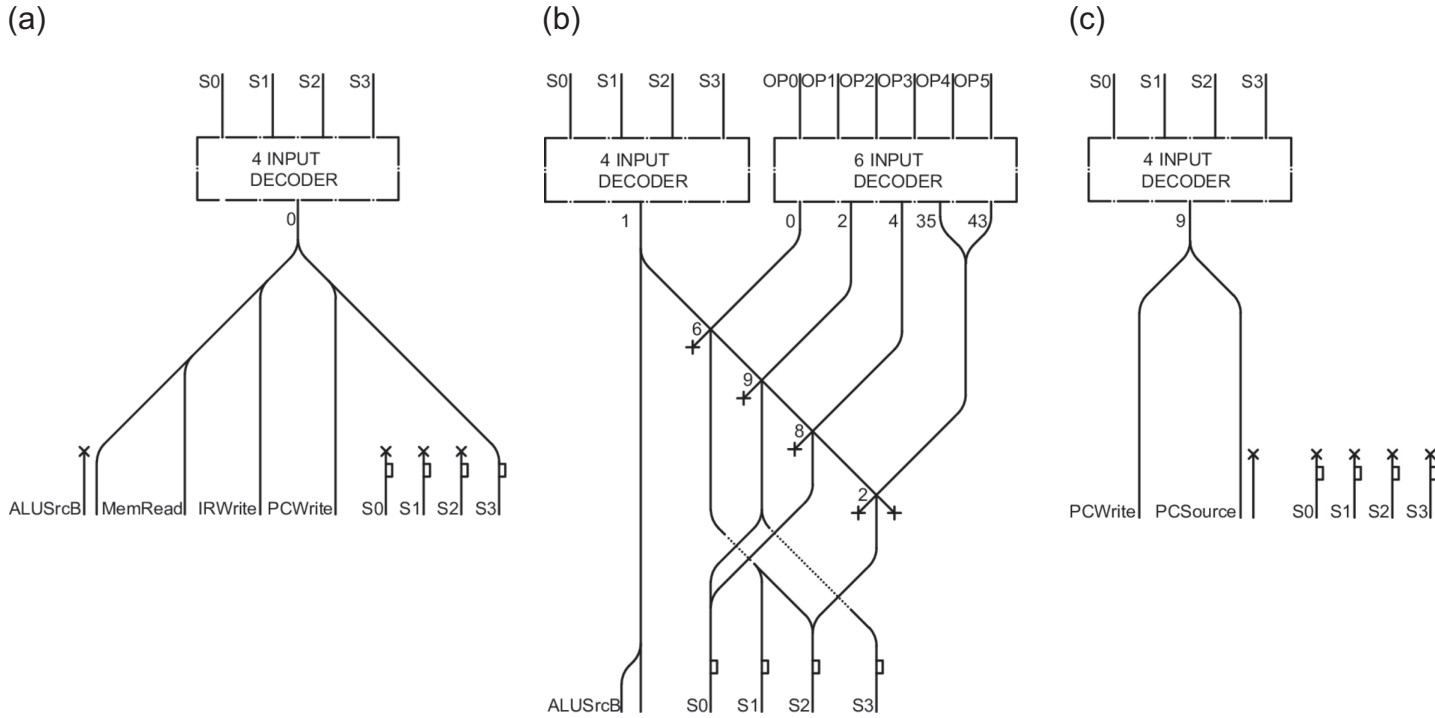

**Fig 15. Fetch phase (a) decode phase (b) jump phase (c).**

The circuit diagram illustrating the fetch, decode, and jump phases, selected as an example operation, is now presented. It is worth noting that the 4 inputs to the *decoder* remain constant throughout all phases: division into multiple images has been done only for the purpose of organization and readability.

## Fetch

The first operation to be performed, regardless of the line of code read, is the fetch in Fig 15(a). This operation scans the value in the *instruction register* and calculates the next Program Counter value, adding the value 4 to it. The fetch code is 0, so the various outputs it needs will be generated from the corresponding *decoder* value. In the fetch phase there is no need to know the code of the current operation, in fact *OP0*, *OP1*, *OP2*, *OP3*, *OP4*, *OP5* are ignored and not reported in the image. During the fetch step, *ALUSrcB* is set to 01, while *MemRead*, *IRWrite*, and *PCWrite* are set to 1, as outlined in Fig 14.

## Decode

After the fetch phase, the decode phase (indicated by a value of 1 in Fig 14) starts. During this phase, the states *S0*, *S1*, *S2*, and *S3* will all be set to 0001. The corresponding LM circuit is depicted in Fig 15(b). In the multi-cycle datapath, the only output to be set is *AluSrcB*, and both their bits must be set to 1. Clearly, subsequent operations require transitioning to distinct states. This necessitates the LM present at output 1 of the 4-bit *decoder* to interact with the outputs of the 6-bit *decoder*, which delineates all possible operations in the multi-cycle datapath. In this example, only states 2, 6, 8, and 9 are used, as outlined in Table 2. For an exhaustive list of possible operations and their respective states, we refer the reader to [28]. The values of *S0*,

**Table 2. Coding of operations and respective state.**

| Op[5-0] | Op type | Next state |
|---|---|---|
| 35, 43 | LW, SW | 2 |
| 0 | R-type | 6 |
| 4 | BEQ | 8 |
| 2 | Jump | 9 |

*S1*, *S2*, and *S3* are set by taking the first 6 bits of the operation (Op[5-0]). These values represent the binary encoding of the next state to move to.

## Jump

The scheme for the jump instruction in Fig 15(c), with state code 9, is then proposed. Considering the state diagram, it is easy to see that the operation is a trivial translation from diagram to circuit. To proceed, three pieces of information are required: the state code (in our example, the value is 9), used to select the channel in the 4-bit *decoder*; the specific control outputs to be configured (in this case, setting *PC source* to 01 and *PC write* to 1); the subsequent state (in this case, 0, indicating readiness to fetch the next instruction). The 6 bits of *OP* are not needed for this step.

By following a similar logic, it is then possible to wire up every single operation in the state diagram described in [28] (chapter 5). Some operations, such as LW load words or SW store words, will require more than one step, but the general structure remains the same.

The *control logic* consists only of *decoders*, which possess good scalability and little waste of resources. Four sensors are needed for channels *S0*, *S1*, *S2*, *S3*, to store the previous state and use it in the next cycle, and these could be an expensive resource. Moreover, in some cases, such as the fetch phase, the LM has to be routed many times, which could be costly.

Being formed by a 5 input *decoder* side by side, with a 6 input *decoder*, the *control logic* will require about 2m x 3m x 0.03m. By considering the circuits arranged in 32 layers, the size results approximately 0.5m x 0.5m x 1m.

## Discussion

Each main component of the multi-cycle datapath in Fig 2 has been, at this point, schematized and analyzed. Other components, including those denoted by *PC*, *Memory Data Register*, *A*, *B*, and *ALUOut*, are standard 32-bit registers, alredy discussed in the previous sections. Among these, only *PC* has an input control signal, specifically its write signal, while for the others the write input is always asserted.

The proposed schemes turn out to be significantly more compact than corresponding schemes implemented by directly combining LM logic gates, like those proposed in [22]. For instance, a three-input *decoder* using classical methods requires eight three-input *AND* gates (or sixteen standard gates), three *NOT* gates (each with an associated syringe for LM creation), and approximately twenty nodes for routing. As illustrated in Fig 6(b), in the proposed scheme the number of gates is reduced to seven, only one syringe is needed for LM generation, and routing is eliminated. This reduction in gates and sensors becomes even more evident as the number of inputs increases, and it can be observed on all the proposed devices.

Some specific issues are also present in some components. Certain components, such as the *register file*, *ALU*, and *control logic*, feature dashed lines. This indicates that they cannot be arranged entirely on a single layer, necessitating an increase in depth, albeit very limited. In

fact, this situation is different from stratification, where the components' patterns change entirely. As previously discussed, the depth required for each layer is not related to the size of LMs, but it depends on the size of the syringes. Therefore, creating two overlapped paths for LMs flowing does not increase this value.

Regarding writing to the *register file*, in Fig 10(a), the *RD* data is passed directly to all inputs since the target register is unknown at this stage. Although graphically these may appear as simple lines, they present a significant challenge: extensive routing is required. Specifically, the 32 bits of data to be written must be routed to each of the 32 different registers. This requires 32 routing mechanisms, each with 32 LMs, even if no stratification is performed.

In the 32 bit *ALU* (Fig 13), layering is not possible, because each *ALU* requires input from the previous one in order to perform its computations. One potential solution is to use a carry lookahead mechanism to anticipate the carry, or to employ sensors that detect the passage of an LM in the previous circuit and can stop the incoming LMs by means of electromagnets, as suggested in [21], so that the computation begins only when all the information is available. While this approach addresses the height issue, it does not resolve the timing problem.

As for the previous case, also in this component, there is an interesting difference compared to classical circuits. In traditional designs, the result of the *Sum* from *ALU31* is connected to the *Cin* of *ALU0* to perform SLT (Set on Less Than) operations. This configuration would have required, in a direct implementation, sending an LM back to *ALU0*, thus resulting in an inefficient implementation. On the contrary, in our proposed approach the *Less* input signal in *ALU0* is always set to 0, just as in all other *ALUs*. In *ALU31*, we execute an *AND* operation among the value of *Sum* and the two bits of *OP*, because it only needs to go to *R0* if the chosen operation is *Less*, which has code 11. The joining with *R0* will never result in an uncontrolled collision, because if there is an LM in *R0* coming from *ALU0*, then the *Less* operation is not executed, and therefore no LMs will come from *ALU31*. If, on the other hand, an LM arrives from this *ALU*, the the operation being exectued is a *Less* operation, and therefore the signal coming from *ALU0* will necessarily be a 0.

The *control logic* will have to be physically placed higher than almost any of the components of the multi-cycle datapath, since these components use the outputs of the *control logic* as inputs. However, the *control logic* itself uses the first six bits of the operation as inputs, which are not immediately available. Unlike other components, the *control logic* relies on the previous state of the system and has outputs that need to be distributed throughout the circuit rather than directed to a single lower point. To efficiently manage the positioning and outputs of the *control logic*, each LM output could be handled with a sensor connected to the appropriate point on the multi-cycle datapath. To minimize the number of sensors required, it is advisable to physically positioning the *control logic* in a higher area, so that only six sensors are needed for the *OP* bits.

We conclude this section by discussing the space and time requirements for the proposed solution. To determine the computation time of the multi-cycle datapath, it is first necessary to estimate its size, using the data collected so far. Considering the estimated sizes of the various components, a complete unstratified circuit measures approximately 140m x 200m. The upper bound of the longest path an LM would travel is therefore 340m. In contrast, for circuits structured in 32 layers, the longest path is reduced to 37m, and with 1024 layers, it is further reduced to 32m.

Table 3 shows the dimensional data for the various components. All units are in meters.

Table 4 shows the approximate travel times for each required component. Each of these is an upper bound, having been calculated as the sum of the height and width measurements and based on the minimum speed of 0.29 m/s for the LMs [1]. Recall that three to five operations are required to perform a datapath operation. Without layering the time to perform one

**Table 3. Size of components (meters).**

| component | 1 layer | 32 layers | 1024 layers |
|---|---|---|---|
| 32 input multiplexer with 32 bits each | 50 x 50 x 0.03 | 1.8 x 1.8 x 1 | 0.25 x 0.25 x 30 |
| 6 input decoder | 1.6 x 1.6 x 0.03 | 0.25 x 0.25 x 1 | 0.25 x 0.25 x 1 |
| register file | 50 x 200 x 0.03 | 4 x 6.5 x 1 | 2 x 0.5 x 30 |
| 32 bit ALU | 20 x 7 x 0.03 | 20 x 7 x 0.03 | 20 x 7 x 0.03 |
| control logic | 2 x 3 x 0.03 | 0.5 x 0.5 x 1 | 0.5 x 0.5 x 1 |
| datapath | 140 x 200 x 0.03 | 30 x 7 x 1 | 25 x 7 x 30 |

**Table 4. Computation time for components with the minimu speed of 0.29 m/s (minutes).**

| component | 1 layer | 32 layers | 1024 layers |
|---|---|---|---|
| 32 input multiplexer with 32 bits each | 5:45 | 0:12 | 0:02 |
| 6 input decoder | 0:11 | 0:02 | 0:02 |
| register file | 14:20 | 0:36 | 0:08 |
| 32 bit ALU | 1:30 | 1:30 | 1:30 |
| control logic | 0:17 | 0:04 | 0:04 |
| datapath | 20:00 | 2:00 | 1:50 |

operation is 1170 seconds (approximately 20 minutes). With 32 layers, this time is reduced to 127 seconds (2 minutes), while considering http://journals.plos.org/plosone/s/latexschemes with 1024 layers the time is further reduced to 110 seconds (1:50 minutes).

Although the change from 32 to 1024 layers makes a substantial difference for some specific components, it does not bring significant advantages when considering the complete datapath. The minimal difference in magnitude is primarily due to the ALU, which cannot be organized into 1024 layers, as discussed above and shown in Table 3. Consequently, the ALU has a fixed size of 20m x 7m, values that set a lower bound for the dimensions of height and width. Nonetheless, authors in [21] pointed out that empirical testing has demonstrated that manipulable LMs can be formed as small as 1.0 μl, still maintaining reliability. This indicates that the devices could be scaled down accordingly, obtaining circuits whose dimensions could be significantly smaller than those mentioned previously.

A further observation regarding the computation times obtained is warranted. It should be strongly considered that if higher speeds are handled correctly, calculation times will be reduced.

As previously mentioned in this paper and discussed in detail in [14], it is possible to bring LMs to speeds of 0.6 m/s without encountering problems. Since this speed is twice that on which the times in Table 4 were calculated, the computation times could potentially be halved.

Now that the various computation times have been presented, a technical issue can be addressed: the coating of LMs wears off over time and needs to be preserved. In [21], it is stated that a typical Ni/UHDPE coating weighs 2.5 mg. The evaporation rate for this coating is calculated to be 0.0998 mg/min. This implies that an LM coated in this material can last over 25 minutes, which is more than sufficient for our purposes, even when considering LMs moving at the minimum speed of 0.29 m/s. Additionally, it should be noted that many LMs are created and destroyed during the computation, making it rare for any single LM to traverse the entire datapath.

Although it has yet to be assessed how LMs react and behave upon partial evaporation of the coating, the latest observations assure us that there is sufficient room for maneuver.

Therefore, it is feasible to consider reintegration of the coating material after each complete operation.

If needed, another possible solution would still be to coat the LMs multiple times during the same whole operation in strategic areas of the circuit, such as after the 32-bit register *Memory* or before insertion into the *ALU*. This approach opens up the possibility of creating a pipelined datapath that divides the operations into sub-phases and reloads the coating after each phase. However, this option involves a completely different functioning and circuit implementation, which is beyond the scope of this work.

## Conclusions

We proposed a theoretical implementation of a complete datapath based on Liquid Marbles, liquid droplets enveloped by a layer of hydrophobic particles on the surface that encapsulate the droplet and keep it spherical.

We started by considering *multiplexers* and *decoders*, which are the basis to many other components. These facilitated the schematization of sequential circuits of increasing complexity, such as *D latch*, *D flip-flop*, and *register file*. Subsequently, we focused on the computational aspect of the datapath, that is the design and implementation of the *ALU* and its different versions. Finally, the *control unit* was considered, proposing the implementation of some key operations. Each circuit has been tested for correctness by means of a Billiard Ball Model simulator [23].

For each circuit, particular emphasis was placed on estimating the physical dimensions of the resulting circuits. Efforts were directed towards minimizing these dimensions. To this end, the option of stratification was proposed and analyzed for each circuit where it proved viable. Indeed, a challenge concerning both height and width emerged, potentially leading to increased construction costs and computation time. Through the implementation of circuit stratification, it was possible to reduce the physical size of the circuits and, as a consequence, it was possible to reduce by 8 times the estimated time to perform one step of an operation.

Given the lack of a physical implementation, some issues remain open that could be addressed in future advances. It would be interesting to find alternative solutions to the use of sensors, so as to simplify sequential devices and *control logic*. Possible multi-cycle datapath exceptions are also to be handled, using the *R* value of register reset (Fig 8(b)) and the *ALU32* Overflow (Fig 12(b)). These two values are already present in the multi-cycle datapath, working but not yet used effectively. Another crucial problem to be addressed is the proper synchronization of each component, which could be solved, for instance, by exploiting the electromagnets.

## Author Contributions

**Conceptualization:** Sandro Erba.

**Formal analysis:** Sandro Erba, Luca Cavenaghi, Claudio Zandron.

**Funding acquisition:** Claudio Zandron.

**Investigation:** Sandro Erba, Luca Cavenaghi, Claudio Zandron.

**Methodology:** Sandro Erba, Luca Cavenaghi, Claudio Zandron.

**Supervision:** Claudio Zandron.

**Validation:** Sandro Erba, Luca Cavenaghi, Claudio Zandron.

**Visualization:** Sandro Erba, Luca Cavenaghi, Claudio Zandron.

**Writing – original draft:** Sandro Erba, Luca Cavenaghi, Claudio Zandron.

**Writing – review & editing:** Sandro Erba, Luca Cavenaghi, Claudio Zandron.

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
