## [Decision Letter · Decision Letter 0]

16 Jun 2024

PONE-D-24-22111Implementing a Multi-cycle Datapath with Liquid MarblesPLOS ONE

Dear Dr. Zandron,

Thank you for submitting your manuscript to PLOS ONE. After careful consideration, we feel that it has merit but does not fully meet PLOS ONE’s publication criteria as it currently stands. Therefore, we invite you to submit a revised version of the manuscript that addresses the points raised during the review process.

We look forward to receiving your revised manuscript.

Kind regards,

Libo Huang

Academic Editor

PLOS ONE

Journal Requirements:

"Universita' Milano-Bicocca, Italy

Project: 2022-ATEQC-0046

"Fondo Ateneo quota Competitiva""

4. Thank you for stating the following in your Competing Interests section: "NO authors have competing interests"

5. We note that your Data Availability Statement is currently as follows: "All relevant data are within the manuscript and its Supporting Information files."

**Additional Editor Comments:**

Please address reviewers' concerns.

Reviewers' comments:

Reviewer's Responses to Questions

**Comments to the Author**

1. Is the manuscript technically sound, and do the data support the conclusions?

Reviewer #1: No

Reviewer #2: Yes

2. Has the statistical analysis been performed appropriately and rigorously? 

Reviewer #1: N/A

Reviewer #2: N/A

3. Have the authors made all data underlying the findings in their manuscript fully available?

Reviewer #1: No

Reviewer #2: Yes

4. Is the manuscript presented in an intelligible fashion and written in standard English?

Reviewer #1: Yes

Reviewer #2: Yes

5. Review Comments to the Author

Reviewer #1: Report on Implementing a Multi-cycle Datapath with Liquid Marbles

by Sandro Erba et al.,

Submitted to Plos one

Summary

Ebra et al. propose the developed version of logic circuits applicable to LMs. They built the scheme of the logic circuits and claimed the model can be used based on LMs, which is not experimentally confirmed but Billiard Ball Model (BBM) simulation.

Recommendation

I recommend that this work not be published in Plos One at this stage due to the insufficient evidence supporting the central claims.

This work explains the correlation between LM and proposed logic circuits with the following:

(i) BBM can model LM + (ii) Model applicability is confirmed by BBM simulation, not LM experiment.

This logic lacks a direct correlation between the LM and the proposed circuit. Actually, LM is a nonsticking “liquid” droplet. While LM is quasielastic, it sways, viscous dissipates, highly deforms, and splits/merges dynamically. Thus, BBM simulation cannot explain the LM behavior perfectly as long as the authors show a critical experimental condition that matches the simulation.

To satisfy this, the LM experiment should have appeared in this work. At least, the authors should demonstrate the logic circuits that newly appeared in this paper, not by previous works, which is absent in the current manuscript.

Thus, I feel this work is an empty theory at this stage.

Major comments:

1. Setups of the LM logic circuits are not illustrated in detail. For example, where is the input or output in Figure 1?

2. Is it necessary to use LM in the proposed logic circuits? If the answer is no, the proposed logic circuits must be new for the BBM system to keep the novelty of this work.

Reviewer #2: The presents a nice idea, however, it definitely needs a revision.

Remarks.

1. Liquid marbles are not stable. The are connected to atmosphere and they evaporate. This will impact the suggested computation via liquid marbles.

2. What is the estimated single computation time?

3. Elastic properties of liquid marbles should be considered, see:

Pogreb, R., Balter, R. et al. Elastic properties of liquid marbles. Colloid Polym Sci 293, 2157–2164 (2015).

6. PLOS authors have the option to publish the peer review history of their article (what does this mean?). If published, this will include your full peer review and any attached files.

Reviewer #1: No

Reviewer #2: No

---

## [Author Response · Author response to Decision Letter 0]

31 Jul 2024

We thank the reviewers for highlighting the issues in the paper and providing valuable suggestions. We have made an effort to better explain our choices and the ideas behind the proposed circuits. 

We carefully considered all the requests and suggestions (both from the reviewers and the editor) and we tried to address them as satisfactorily as possible.

---

## [Decision Letter · Decision Letter 1]

6 Aug 2024

Implementing a Multi-cycle Datapath with Liquid Marbles

PONE-D-24-22111R1

Dear Dr. Zandron,

We’re pleased to inform you that your manuscript has been judged scientifically suitable for publication and will be formally accepted for publication once it meets all outstanding technical requirements.

Kind regards,

Libo Huang

Academic Editor

PLOS ONE

Additional Editor Comments (optional):

Reviewers' comments:

Reviewer's Responses to Questions

**Comments to the Author**

1. If the authors have adequately addressed your comments raised in a previous round of review and you feel that this manuscript is now acceptable for publication, you may indicate that here to bypass the “Comments to the Author” section, enter your conflict of interest statement in the “Confidential to Editor” section, and submit your "Accept" recommendation.

Reviewer #1: All comments have been addressed

Reviewer #2: All comments have been addressed

2. Is the manuscript technically sound, and do the data support the conclusions?

Reviewer #1: No

Reviewer #2: Yes

3. Has the statistical analysis been performed appropriately and rigorously? 

Reviewer #1: No

Reviewer #2: Yes

4. Have the authors made all data underlying the findings in their manuscript fully available?

Reviewer #1: (No Response)

Reviewer #2: Yes

5. Is the manuscript presented in an intelligible fashion and written in standard English?

Reviewer #1: Yes

Reviewer #2: Yes

6. Review Comments to the Author

Reviewer #1: (No Response)

Reviewer #2: The paper is publishable. The author carefully addressed the remarks. The results are novel. The paper is well-written and clearly organized. General impression from the paper is very good.

7. PLOS authors have the option to publish the peer review history of their article (what does this mean?). If published, this will include your full peer review and any attached files.

Reviewer #1: No

Reviewer #2: No

---

## [Editor Report · Acceptance letter]

9 Aug 2024

PONE-D-24-22111R1 

PLOS ONE

Dear Dr. Zandron, 

I'm pleased to inform you that your manuscript has been deemed suitable for publication in PLOS ONE. Congratulations! Your manuscript is now being handed over to our production team.

Kind regards, 

on behalf of

Prof. Libo Huang 

Academic Editor

PLOS ONE